# Pharmacokinetic Drug–Drug Interactions among Antiepileptic Drugs, Including CBD, Drugs Used to Treat COVID-19 and Nutrients

**DOI:** 10.3390/ijms22179582

**Published:** 2021-09-03

**Authors:** Marta Karaźniewicz-Łada, Anna K. Główka, Aniceta A. Mikulska, Franciszek K. Główka

**Affiliations:** 1Department of Physical Pharmacy and Pharmacokinetics, Poznan University of Medical Sciences, 60-781 Poznań, Poland; mkaraz@ump.edu.pl (M.K.-Ł.); amikulska@ump.edu.pl (A.A.M.); 2Department of Bromatology, Poznan University of Medical Sciences, 60-354 Poznań, Poland; aglowka@ump.edu.pl

**Keywords:** AEDs, cannabidiol, pharmacokinetics, biotransformation, therapeutic levels, clinical interactions

## Abstract

Anti-epileptic drugs (AEDs) are an important group of drugs of several generations, ranging from the oldest phenobarbital (1912) to the most recent cenobamate (2019). Cannabidiol (CBD) is increasingly used to treat epilepsy. The outbreak of the SARS-CoV-2 pandemic in 2019 created new challenges in the effective treatment of epilepsy in COVID-19 patients. The purpose of this review is to present data from the last few years on drug–drug interactions among of AEDs, as well as AEDs with other drugs, nutrients and food. Literature data was collected mainly in PubMed, as well as google base. The most important pharmacokinetic parameters of the chosen 29 AEDs, mechanism of action and clinical application, as well as their biotransformation, are presented. We pay a special attention to the new potential interactions of the applied first-generation AEDs (carbamazepine, oxcarbazepine, phenytoin, phenobarbital and primidone), on decreased concentration of some medications (atazanavir and remdesivir), or their compositions (darunavir/cobicistat and lopinavir/ritonavir) used in the treatment of COVID-19 patients. CBD interactions with AEDs are clearly defined. In addition, nutrients, as well as diet, cause changes in pharmacokinetics of some AEDs. The understanding of the pharmacokinetic interactions of the AEDs seems to be important in effective management of epilepsy.

## 1. Introduction

Epilepsy is a chronic brain disease that affects 65 million people worldwide and of all ages. The incidence rate is highest in young children under 2 years of age and adults aged 65 or older. About 50–60 percent of children with seizures eventually grows out of them and never experience seizures as an adult. It is the leading neurological cause of loss of quality-adjusted life years. Appropriate treatment can only be carried out after an accurate diagnosis of epilepsy, which is conducted mainly on clinical trials. Electroencephalography and neuroimaging based first of all on magnetic resonance imaging are supporting diagnostic investigations. Antiepileptic drugs (AEDs) are the pivotal treatment for epilepsy and achieve seizure-free treatment in about two-thirds of patients. The medications should be selected taking into account the type of epilepsy (generalized or focal) and concomitant medical and mental illnesses of the patient. Patients with drug-resistant epilepsy require a special approach [1,2]. Women require special attention due to possible hormonal changes, which may be caused by taking anti-epileptic drugs. Effective contraception and safe anti-epileptic therapy during pregnancy are especially important due to the teratogenicity of AEDs. In addition, women with epilepsy have a greater risk of bone loss because AED treatment is associated with a reduction in bone mineral density. Therefore, calcium and vitamin D supplements may be required during the menopause [3].

Combinations of AEDs are used frequently in patients not responding to monotherapy. Polytherapy is common especially in patients with refractory epilepsy and elderly people. In such situations clinically important interactions are common [4]. Other groups, such as patients treated with anti-cancer drugs, immunosuppressants and HIV-positive patients or with psychiatric impairments, often take multidrug therapies, including AEDs. Moreover, other common disorders, such as neuropathic pain and migraine, are often treated with AEDs [5].

The most clinically important AEDs interactions are pharmacokinetic, involving changes in absorption, protein binding, metabolism, or excretion. Many AEDs, especially older generations, inhibit hepatic P450 enzymes (valproic acid). The group of the P450 inducers involves carbamazepine, phenytoin, phenobarbital and primidone. AEDs that are enzyme-inducers affect endogenous biochemical pathways, including metabolism of sex hormones, vitamin D hoemostasis, or bone metabolism [6]. Although newer AEDs have better pharmacokinetic properties and a lower risk of interactions than older ones, they are often used in combination with older drugs.

Cannabidiol belongs to the newer anti-seizure medications and its efficacy in the treatment of epileptic seizures, especially in children, has been confirmed in several studies. As an inducer of CYP1A2 and CYP2B6 and an inhibitor of uridine-glucuronosyltransferases, CYP2C8, CYP2C9 and CYP2C19, it could potentially interact with co-administered AEDs and other drugs [7].

Currently, potential drug–drug interactions between AEDs and anti-COVID-19 therapies may be clinically significant challenges. The interactions with antiviral and immune medicines may lead to toxicity, treatment failure, increasing adverse effects and seizure frequency [8]. Moreover, clinicians should be vigilant towards interactions between AEDs and COVID-19 vaccines, because vaccines could modify the metabolism of the drug by changed expression of CYP450 enzymes due to modulation of immune system [9].

Dietary therapy with the ketogenic diet or the modified Atkins diet based on low-carbohydrate, high-fat meals allow at least a 50% seizure reduction and improved cognition and mood in epileptic patients. It is suggested that the mechanism underlying efficacy of dietary therapy involves the inhibition of the α-amino-3-hydroxy-5-methyl-4-isoxazolepropionic acid receptor by decanoic acid, a component of triglycerides. Moreover, it was suggested that ketone bodies control seizures [1]. It should be borne in mind that foods and nutrients may influence the pharmacokinetics of AEDs. On the other hand, AEDs may cause changes in the nutritional status of patients, leading to deficiency of vitamins and mineral components and, consequently, changing response to a given drug [10].

Therapeutic drug monitoring (TDM) allows to manage fluctuations in serum/plasma drug concentrations and changed clinical response resulting from drug–drug interactions. It can be helpful in enabling the definition of individual therapeutic concentrations and to control dosing to minimize drug interactions and prevent loss of efficacy or toxicity [11].

In the present review, we focus on clinically relevant interactions between AEDs and AEDs in combination with other important drugs, including anti-COVID-19 medicines. Moreover, interactions of AEDs with food and nutrients are analyzed, including deficiencies in vitamins and micro- and macro-elements. This article presents recent advances regarding drug interactions, including new AEDs such as cannabidiol and cenobamate, that have not been described in previous reviews.

## 2. Cannabidiol

Cannabidiol (CBD) is a non-psychotropic phytocannabinoid obtained from *Cannabis sativa* L. [12]. CBD influences the activity of serotonin, opioid and non-endocannabinoid G-protein coupled receptors (GPCR) [13]. The drug shows immunosuppressive and neuroprotective properties resulting from anti-inflammatory action [14]. Therefore, it is used in neurodegenerative diseases, including multiple sclerosis, Parkinson’s disease, Huntington’s disease, or Alzheimer’s disease, but also in the treatment of pain, cancer, emesis, anorexia, irritable bowel syndrome, infectious diseases, glaucoma and central nervous system disorders [15]. Interest is also aroused by the antidepressant effect of CBD and reducing forms of anxiety, such as obsessive-compulsive disorder and post-traumatic stress disorder [16]. Research on CBD has provided significant information on its anticonvulsant activity, as evidenced by the effectiveness of the treatment of epileptic seizures, especially in children [17]. The mechanisms of the anticonvulsant activity of CBD, although known to be multifactorial, are still only partially understood. They are linked to the blockage of the T-type Ca^2+^ channels and opioid receptors, as well as the agonist action towards GPCR [7]. The pharmacokinetics of CBD exhibits wide inter-patient variability. C_max_ and AUC are dose-dependent and half-life depends on the dose and route of administration (Table 1). Absorption, plasma protein binding, metabolism and excretion are reduced in children, compared to adults [18].

The available data on the interaction of CBD with other antiepileptic drugs indicate harmless side effects that can be eliminated by reducing the dose of CBD or drugs commonly used in the treatment of epilepsy. Adverse events during concomitant use of CBD with valproate relate to abnormal liver function; therefore, it is very important to monitor these compounds along with their metabolites in the serum [7]. Co-administration of CBD with clobazam, which is a competitive inhibitor of CYP3A4, CYP2C19 and CYP2B6, leads to increases in C_max_ of 73% and AUC of 47% for CBD and its active metabolite 7-hydroxy-CBD. Moreover, concentrations of clobazam and its active metabolite increase by 60% and 3–5 fold, respectively [74]. CBD increases concentrations of eslicarbazepine, rufinamide, stiripentol and topiramate (Table 2). CYP3A4 and CYP2C19 inducers, including carbamazepine, topiramate and phenobarbital, may decrease CBD bioavailability and effectiveness. CYP2C19 inhibitors, including fluvoxamine, fluoxetine, proton pump inhibitors, cimetidine, ketoconazole, clopidogrel and efavirenz, could increase CBD bioavailability and, possibly, decrease CBD effectiveness [74].

During the COVID-19 pandemic, patients treated with antiepileptic drugs, as well as antiviral medications, are at risk of drug interactions, especially when the drugs are metabolized by the same cytochrome P-450 enzymes. CBD, as an inhibitor of CYP2C8 and CYP3A4, may cause an increased plasma concentration of darunavir/cobicistat and lopinavir/ritonavir compositions, as well as hydroxychloroquine. In contrast, its concentration might be increased when co-administered with ritonavir, a potent inhibitor of CYP3A/CYP2D6 (Table 3).

## 3. Alkyl-Carbamates: Cenobamate and Felbamate

### 3.1. Cenobamate

Cenobamate (CNB), a new antiepileptic medication, was approved by the FDA in 2019 to treat uncontrolled partial-onset seizures in adult epileptic patients. It inhibits sodium channels and positively modulates GABAA ion channels to prevent partial onset seizures.

After single oral doses of CNB in the range of 5–750 mg (0.0125–1.88 times the maximum recommended dose), AUC increased more than proportionally [31,77]. However, after multiple dosing from 50 to 500 mg/day, the AUC increased in a dose-proportionate manner [32,33]. C_max_ increased in a proportional way with the dose. The steady state concentration was achieved after nearly 2 weeks. Up to now, there is no observed influence of age, sex, or race/ethnicity on clinically important pharmacokinetic parameters of CNB. In patients with mild (CL_cr_ 60-less than 90 mL/min) and moderate (CL_cr_ 30 to less than 60 mL/min) renal impairment, following single oral administration of 200 mg dose tablets, a 1.4–1.5 increase in AUC was observed, if compared with healthy volunteers [32,33]. The pharmacokinetic parameters of cenobamate are summarized in Table 1.

A significant increase in C_max_ and AUC of phenytoin (PHT), by 70% and 84%, respectively, and the same parameters of phenobarbital (PHB), by 34% and 37%, were observed after concomitant multiple administrations of 200 mg CNB. In the case of carbamazepine (CBZ), administration of CNB decreased C_max_ and AUC, each by 23%. A similar situation was viewed from population pharmacokinetics of lamotrigine (LTG), in which concentrations were smaller, by about 23%. The not-clinically significant decrease by 4–13% in levetiracetam (LEV) concentrations was reported. No clinically significant differences in the pharmacokinetic parameters were observed in the case of valproic acid (VPA) and lacosamide (LCM) concomitantly administered with CNB. Moreover, CNB increased the plasma concentration of N-CLB, the active metabolite of clobazam. This may increase the risk and/or severity of side effects, such as drowsiness, tiredness, drooling, constipation and breathing difficulties [31].

Moreover, CNB decreases C_max_ and AUC of bupropion-CYP2B6, by 23% and 39%, respectively, midazolam-CYP3A substrate, by 61% and 72%, respectively, and oral contraceptives. The pharmacokinetics parameters of the omeprazole-CYP2C19 substrate were increased, but no significant changes in pharmacokinetic parameters were noted in the case of warfarin (also a CYP2C19 substrate) [31].

Decreased C_max_ and AUC of CNB, by 27% each, were found when CNB was administered concomitantly with PHT. However, no significant impact of repeated doses of VPA, PHB, or CBZ on CNB was observed [31].

The high-fat meal diet did not influence the pharmacokinetics of CNB; in addition, there was no observed interaction with alcohol when concomitantly administered with CNB [31,77].

There are potential interactions between CNB and medications used in the treatment of the patients with COVID-19. CNB, as an inducer of CYP3A4, can affect concentrations of hydroxychloroquine/chloroquine; then, an increase in the drug dose is required [75]. However, WHO halted the trial study on hydroxychloroquine and lopinavir/ritonavir, because of little or no reduction in the mortality of hospitalized COVID-19 patients, when compared to standard of care [78]. CNB decreased concentrations of the medications atazanavir, darunavir/cobicistat and lopinavir/ritonavir used in the treatment of COVID-19 [76] (Table 3). CNB belongs to the group of AEDs with potential interaction, which requires dose adjustment.

### 3.2. Felbamate

Felbamate (FBM) was synthesized and developed in the 1950s and, 30 years later, its antiseizure profile was defined. It enhanced GABAergic neurotransmission at the GABAA receptor, similarly to PHB, as well as modulated voltage-dependent sodium channels. In 1993, it was approved for the treatment of focal seizures in adults and Lennox–Gastaut syndrome [79].

FBM represents linear pharmacokinetics in therapeutics doses of 800–1200 mg, three times daily. Its therapeutic range is 30–60 mg/L. Biological half-life in adults is 10–30 h; therefore, it takes 3–5 days to obtain steady state. Unchanged FBM is excreted by the kidneys, the rest is metabolized in the liver by the CYPs (Table 1). Its reactive atropaldehyde metabolite is a cause of hepatotoxicity and aplastic anemia in a few patients; therefore, its clinical use was declined [19,29,80].

FBM inhibits the activity of CYP3A4; therefore, important pharmacokinetic interactions with other inducers of the family isoenzymes are observed (Table 2). FBM decreases the level of CBZ, but increases the level of its active metabolite CBZE, as well as PHB, PHT and VPA. These interactions are important from a clinical point of view because of the narrow therapeutic range, as well as non-linear pharmacokinetics of PHT. FBM interacts with other therapeutic classes and decreases the level of hormonal contraceptives. There are some pharmacokinetic interactions affecting FBM. PHT and CBZ increase the elimination of FBM. VPA and gabapentin (GBP) increase FBM plasma concentration but by different mechanisms; the former, VPA, causes it by inhibition of FBM metabolism, while GBP by inhibition of renal elimination. In pediatric patients with epilepsy, the impact of FBM, as well as other AEDs, on pharmacokinetic variability of CLB is extensive and resulting in drug–drug interactions and age [19,29,81]. Blood monitoring of FBM concentrations is recommended because of a variable metabolism and clearance dependent on age [82,83,84]. Noteworthy is also the effect of FBM on body weight loss in patients treated with this drug [10].

Medications used in the treatment of COVID-19, such as atazavir, lopinavir/ritonavir chloroquine and hydrochloroquine, could decrease exposure to FBM, while the drug could decrease the concentration of darunavir/cobicistat (Table 3). Moreover, the drug can be susceptible to interaction with COVID-19 vaccines [9].

## 4. Clobazam

Clobazam (CLB) is a 1,5-benzodiazepine acting as a partial-aminobutyric acid (GABA) receptor agonist. The drug possesses anxiolytic, sedative and anticonvulsant properties. CLB selectively binds to the α2 subunit of the GABA A receptor, which is associated with antiepileptic activity, and, less selectively, binds to the α1 subunit. CLB is approved for the adjunctive treatment of seizures associated with Lennox–Gastaut syndrome in children and adults. Moreover, CLB is used in different types of seizure, such as absence seizures, atonic seizures, complex partial seizures, focal aware or simple partial seizures and tonic–clonic seizures, as well as with alcohol withdrawn [85].

CLB represents linear pharmacokinetics. The therapeutic range for CLB is 0.03–0.30 mg/L and 0.3–3.0 mg/L for N-CLB [19,29,35,86]. The drug undergoes extensive metabolism in the liver to an active metabolite N-desmethylclobazam (N-CLB) and other metabolites (Table 1). The CYP-dependent metabolism of CLB may be subject to drug–drug interactions (Table 2). CYP3A inducing antiepileptic drugs (PHB, PHT and CBZ) and FBM may increase N-CLB levels, while CYP2C19 inducers may increase N-CLB excretion [87]. Stiripentol (STP) is an inhibitor of CLB and N-CLB metabolism and may increase the concentration of N-CLB several times. The most likely mechanism of the interaction is the inhibition of the enzyme CYP2C19 [88]. CBD interacts significantly with CLB. It was found that, increasing the dose of CBD, CLB levels decreased, but serum levels of N-CLB increased, in pediatric and adult patients [23]. The increase in plasma concentration of the active metabolite of CLB was 3.4–5 folds [24].

The CYP and UGTs inhibitors atazanavir, darunavir/cobicistat and lopinavir/ritonavir could potentially increase exposure to CLB (Table 3). Extracorporeal membrane oxygenation (ECMO) used in the management of patients with COVID-19 may potentially affect the pharmacokinetics of highly protein-bound drugs, including CLB, leading to an increase in its pharmacological effect [75].

## 5. Dibenzazepines: Carbamazepine, Eslicarbazepine Acetate and Oxcarbazepine

Carbamazepine (CBZ), eslicarbazepine acetate (ESL) and oxcarbazepine (OXC) structurally belong to the dibenzazepine-carboxamides class. The mechanism of action of these AEDs involves inhibition of the voltage-gated sodium channels, but they also modulate different types of calcium channels [37,89].

### 5.1. Carbamazepine

CBZ is one of the most commonly prescribed anticonvulsants and the oldest antiepileptic drug. It is used for the treatment of focal-onset epilepsy and primary generalized tonic–clonic seizures. Oral absorption of CBZ is variable but is not affected by food. The pharmacokinetics of CBZ is nonlinear due to the induction of its own metabolism. The pharmacokinetic parameters of CBZ are presented in Table 1. The time to reach steady state concentrations takes 3–5 weeks as result of autoinduction [30].

The narrow therapeutic range, nonlinear pharmacokinetics and unpredictable relationship between its dose and plasma levels, as well other factors, such as genetic polymorphisms of metabolizing enzymes, patients’ age and sex, or autoinduction, make TDM a useful tool in CBZ therapy. In practice, both CBZ and CBZE are monitored. The current reference range for CBZ and CBZE in human plasma is 4–12 mg/L and up to 2.3 mg/L, respectively [19,29].

The administration of the VPA and CBZ combination results in increased concentrations of active carbamazepine 10–11 epoxide. PHB, PHT or/and primidone (PRM) decrease levels of CBZ. Drugs that are more rapidly metabolized with CBZ include LTG, PHT and VPA, as well theophylline (Table 2). Other class drugs which rapidly undergo biotransformation with CBZ and, consequently, their concentrations are decreased, include antibiotics (erythromycin), H2 blockers (cimetidine), opioid analgesic (propoxyphene) and calcium channel blockers. CBZ also increases the metabolism of contraceptives compounds and, finally, can reduce their effectiveness. As inducer of CYP450, it accelerates the elimination of many benzodiazepines and decreases their action [30]. CBZ induces the metabolism of warfarin, leading to an increase in the warfarin dose required to keep the International Normalized Ratio (INR) within the therapeutic range [90]. CBZ interacts with apixaban anticoagulant medication, causing sub therapeutic concentration of the drug in patients [91].

Grapefruit juice increases the bioavailability of CBZ by inhibiting CYP3A4 enzymes in the gut wall and in the liver [92]. Similarly, pomegranate juice, as an inhibitor of cytochrome CYP3A4, significantly increases the AUC of orally administered CBZ in rats [93].

A ketogenic diet causes the decrease in the serum concentration of CBZ in young patients (median, 2.91 years), but not significantly [94]. In infants, the ketogenic diet significantly decreased CBZ concentration in 44% during the effective treatment of early onset epilepsy which includes high doses of PHT or CBZ [95].

CBZ decreases significantly concentrations of atazanavir, darunavir/cobicistat, lopinavir/ritonavir, remdesivir, chloroquine and hydrochloroqiune. However, the interaction of CBZ with atazanavir, darunavir/cobicistat and lopinavir/ritonavir may increase the concentration of CBZ and cause toxicity (Table 3). The reason for toxicity is the inhibition of CYP3A4. Therefore, AEDs should not be co-administered with these anti-COVID-19 medications [76].

### 5.2. Eslicarbazepine

Eslicarbazepine acetate is a prodrug of eslicarbazepine (ESL). ESL, like CBZ and OXC, belongs to the family of the dibenzazepine carboxyamides. It is the third generation of the carboxamide family (CBZ, OXC). ESL is a reduced stereoselective S-metabolite (S-licarbazepine) of OXC. It is indicated as either adjunctive or monotherapy treatment for partial seizures in adults and children four years of age and older [96,97].

The mechanism of action of ESL involves blocking voltage-gated sodium channels and stabilizing their inactive state. The drug can be administered as a single daily dose, does not have any significant interactions with other drugs and possesses a favorable safety profile [36,98].

ESL represents a linear relationship between dose and serum concentration at clinically relevant doses of 400–1600 mg/day. ESL reaches steady state in the plasma concentration after 2–4 days. The pharmacokinetics does not depend on age, sex, nor food (Table 1). The reference range for ESL is 3–26 mg/L [86]. ESL, the anticonvulsant metabolite, is subsequently metabolized by conjugation with glucuronic acid via UGT isozymes (1A4, 1A9, 2B4, 2B7 and 2B17) [36].

ESL, like OXC, induces CYP3A4 and UDP-glucuronosyltransferase, but undergoes very little pharmacokinetic drug–drug interaction (Table 2). ESL has an advantage over CBZ and OXC in a smaller number of interactions. However, ESL induces CYP3A4-dependent metabolism of levonorgestrel and ethinylestradiol and reduces the effectiveness of the oral contraceptive hormones. Moreover, it may affect PHT metabolism by inhibition of CYP2C9 and CYP2C19. Conversely, CBZ, PHT, PHB and TPM increase the clearance of ESL and decrease its plasma concentration by about 20–33%. ESL reduces concentration of simvastatin and rosuvastatin by 54% and up to 39%, respectively [29,99,100]. CBD significantly increase levels of ESL in serum [23,25].

ESL shows the same interactions as OXC, which means that it significantly decreases exposure to atazanavir, darunavir/cobicistat, lopinavir/ritonavir and remdesivir (Table 3). Additionally, it may cause QT and PR prolongation on electrocardiogram, if administered together with atazanavir, as well as lopinavir/ritonavir composition [76].

### 5.3. Oxcarbazepine

Oxcarbazepine (OXC), keto analog of CBZ, is an effective anti-epileptic drug used in the treatment of generalized tonic–clonic and partial seizures and trigeminal neuralgia. It is indicated as monotherapy or adjunctive therapy in adults and children 6 years of age and older. It is a structural analog of CBZ, but with different metabolite pathways [101].

OXC, as a prodrug, undergoes biotransformation by cytosolic aryl ketone reductase to the pharmacologically active, stereoselective 10-hydroxy carbamazepine (MHD). The metabolite consists in 80% of (S)-licarbazepine, (eslicarbazepine) and 20% of (R)-licarbazepine. OXC is an active AED, but, because of its extensive metabolism (Table 1), its levels are very low; therefore, only its active monohydroxy metabolite is monitored. The therapeutic range of MHD is in the range of 3–36 mg/L [29,47,86]. The antiseizure action of OXC is exerted mainly by the metabolite MHD. Both OXC and its active metabolite inhibit CYP2C19 and induce CYP3A4/5, which causes metabolic interactions with other drugs [102,103].

The OXC with mild to moderate potential for drug interactions shows relatively few interactions (Table 2). Enzyme-inducing AEDs, such as PHT, PHB or CBZ, may slightly decrease MHD levels. The influence of OXC on other AEDs is, in most cases, not-clinically relevant. OXC increases the concentration of PHT and reduces the trough levels of LTG and topiramate (TPM). In addition, OXC reduces the levels of contraceptives containing ethinyl estradiol and levonorgestrel. Verapamil may moderately reduce MHD levels, but this effect is rather not significant clinically [47].

The therapeutic efficacy of OXC may be decreased when used in combination with hydroxychloroquine or chloroquine. On the other hand, OXC decreases significantly exposure to atazanavir, darunavir/cobicistat, lopinavir/ritonavir, remdesivir, chloroquine and hydrochloroquine [76]. OXC can decrease the serum levels of sofosbuvir [75] (Table 3).

## 6. Ethosuximide

Ethosuximide (ESM) was registered in 1960 in the USA as a racemic mixture of two enantiomers and was approved for the management of absence seizures in patients over 3 years of age. Long therapy studies confirmed the lack of differences of stereoselectivity in metabolism and elimination rates of two enantiomers of ESM [104].

The ESM medication decreases the threshold of brain currents by blocking T-type calcium channels. There is currently no off-label use for ESM; however, there is some evidence that it may have some analgesic effects [38].

ESM generally represents linear pharmacokinetics. However, at higher doses, clearance may be saturable (non-linear). ESM passes into breast milk, resulting in the ratio of the ESM concentration of breast milk vs. plasma of 0.94 ± 0.06. Its protein binding is negligible (Table 1). The concentration of ESM in the saliva is the same as in plasma [39].

The therapeutic range for ESM amounts to 40–100 mg/L. Levels higher than 100 mg/L are usually tolerated without toxicity. Patients with absent seizures may need serum levels above 120 mg/L for seizure control. It is known that levels higher than 100 mg/L are generally tolerated, without side effects [19,29,37,38].

The metabolism of ESM can involve induction or inhibition. ESM is known to interact with antiepileptic inducers such as PHT, CBZ, PHB, PRM and rifampicin, resulting in a decrease in serum ESM levels. ESM may increase PHT levels, but it has no enzyme-inducing properties (Table 2). In contrast, isoniazid and STP may reduce ESM metabolism and clearance. The effect of VPA on ESM concentrations can be variable [19,38].

The last research data presented the possible interaction of ESM metabolized by CYP3A4 and CYP2E1 with COVID-19 vaccines [9]. The CYP and UGTs inhibitors atazanavir, darunavir/cobicistat and lopinavir/ritonavir could potentially increase exposure to ESM (Table 3).

## 7. GABA Analogues

A role of GABA analogues, including gabapentin (GBP), pregabalin (PGB), tiagabine (TGB) and vigabatrin (VGB), is to increase the neurotransmitter GABA in the brain, thus preventing convulsions. GBP, as a precursor of GABA, easily enters the brain and increases brain synaptic GABA concentration. Moreover, it decreases the influx of calcium ions into neurons via a specific subunit of voltage-dependent calcium channels. TGB inhibits neuronal and glial uptake of GABA, while VGB increases the synaptic concentration of the neurotransmitter by inhibition of GABA-transaminase [105]. GBP and PGB are approved for the treatment of neuropathic pain, anxiety and partial seizures. PBG exhibits a linear relationship between given doses and steady-state concentrations, while GBP pharmacokinetics is non-linear as a result of saturable absorption after oral administration. Metabolism of GBP and PGB is negligible and both drugs are renally excreted [41] (Table 1). Therefore, both drugs have a low profile of interaction with other drugs. GBP shows pharmacodynamic synergistic interaction with TPM in the mouse model [106]. Similar effects were observed in the combination of GBP or PGB with LEV and LCM [107]. There is a case report on the interaction between GBP and PHT leading to the increased total and free PHT serum levels [108]. A higher number of side effects was noticed in patients treated with GBP and morphine due to pharmacokinetic interaction. Reduced intestinal motility caused by morphine led to a 44% increase in GBP AUC and 25% increase in the amount of GBP excreted in urine. On the other hand, GBP enhanced the analgesic effect of morphine [109]. A synergistic effect for alleviating pain problems was also reported for the combination of GBP with tramadole and metamizole [110]. Moreover, GBP and PGP interacted additively with naproxen to reverse thermal hyperalgesia associated with peripheral inflammation [111]. Caffeine and sertraline can reduce, while carvedilol and cimetidine augment the anticonvulsant effects of GBP [110].

TGB is useful in the control of partial seizures [67,105]. The drug shows linear pharmacokinetics with a wide distribution throughout the body, high protein binding and extensive metabolism by CYP3A to at least four pharmacologically inactive metabolites [67,68] (Table 1). The TGB clearance can be increased in patients treated with drugs such as CBZ, PHB, PHT or PRM [112]. Increased anticonvulsant activity of TGB, PGB and GBP related to pharmacokinetic interactions with CBD was observed in mice [113].

VGB is a racemic mixture of two enantiomers, the pharmacologically active (+)-*S*-enantiomer and the inactive (−)-*R*-enantiomer. The drug is recommended as adjunctive therapy in resistant epilepsy and for monotherapy in infantile spasms in West syndrome [105]. VGB does not bind to serum proteins and is not metabolized by the liver; thus, it has little interaction with other drugs [73]. After co-administration with PHT or CBZ, it can cause a decrease in concentrations of both drugs and need for dose adjustment in some patients. A similar effect was observed in the case of PRM or PHB but without clinical relevance [114]. The potentiated pharmacodynamic effect was observed when VGB was given with VPA and TGB in mice, however, without changes in plasma concentrations [115,116]. Stereoselective interaction was observed under co-administration of FBM, which increased the plasma concentration and the amount of *S-*(+)-vigabatrin excreted in urine [117]. A protein-rich meal given with VGB caused decreased plasma C_max_ of the drug and prolongation of T_max_ in healthy volunteers [118]. A similar effect was observed when VGB was given to rats fed with infant formula [119]. The effect may be related to a decrease in gastric emptying and/or a direct effect on the proton-coupled amino acid transporter 1 (PAT1) which mediates in the drug absorption [114].

Currently, there is no evidence for any significant drug interactions among GBP, PGB and VGB and medications used to treat COVID-19 patients. TGB, as a substrate for CYP3A4, may be susceptible to the interaction with COVID-19 vaccines due to interferon-gamma production from T-cell responses elicited by COVID-19 vaccines. It was reported that exposure to interferon gamma led to downregulation of CYP1A2 and CYP3A4 in human hepatocytes and lower activities of the CYP enzymes [9]. Moreover, TGB concentration could be increased when co-administered with the CYP and UGTs enzyme inhibitors atazanavir, darunavir/cobicistat and lopinavir/ritonavir (Table 3).

## 8. Lacosamide

Lacosamide (LCM) is indicated for the treatment of focal seizures and as adjunctive therapy for primary generalized tonic–clonic seizures. Its mechanism of action involves enhancing the slow inactivation voltage-gated sodium channels and interaction with the collapsing-response mediator protein-2. Its bioavailability is about 100% and does not depend on food administration (Table 1). No clinically relevant interactions were reported when LCM was co-administered with other AEDs, digoxin, metformin, omeprazole, midazolam, oral contraceptives, or warfarin [43]. In the seizure mice model, LCM in combination with CBZ, PHT, VPA, LTG, TPM, GBP and LEV displayed a synergistic effect. Moreover, coadministration of LCM with TPM led to an increase in plasma levels of LCM, while coadministration of the drug with PHT resulted in an increase in PHT concentrations [120]. The combination of three drugs, including LCM, CBZ and PHB, exerted additive interaction with a slight tendency towards antagonism in the mouse MES-induced seizure model [121]. Another study on the interaction between LCM and second-generation AEDs revealed that the combination of LCM with TPM and PGB was the most beneficial, offering synergistic suppression of tonic–clonic seizure in mice [122]. Enzyme-inducing AEDs may influence LCM pharmacokinetics. Co-administration of CBZ, PHT or PHB resulted in a 15–20% reduction in plasma LCM concentrations. In contrast, LCM decreased concentration of OXC [123]. The activity of LCM might be impaired during therapy used in COVID-19, including hydroxychloroquine, chloroquine and tocilizumab. In contrast, ritonavir, a potent inhibitor of CYP3A/CYP2D6 may increase plasma levels of the drug (Table 3). LCM should be used with caution in COVID-19 patients with cardiac problems, because the drug may prolong the PR interval and cause first-degree atrioventricular block (AV), second-degree and complete AV blocks. In such patients, obtaining ECG is recommended before beginning the treatment with LCM and after obtaining the steady-state LCM concentration [75].

## 9. Lamotrigine

Lamotrigine (LTG) is recommended as adjunctive therapy for focal and generalized tonic–clonic seizures in adults and children, for treatment of bipolar disorder and neuropathic pain and as a first-line drug for the treatment of pregnant women [124]. It undergoes extensive metabolism via uridine 5′-diphosphate glucuronosyltransferase (UGT) isoenzymes UGT 1A4, 1A1 and 2B7 [45] (Table 1). VPA inhibits LTG metabolism and leads to an increase in serum levels while enzyme-inducing anti-epileptics reduces the LTG half-life [124]. Because of its proven synergistic effect, the combination of LTG and VPA is used frequently. Koristkova et al. [125] reported that VPA decreased LTG clearance by 54% in bitherapy and by 21% in triple therapy with CBZ. VPA concentrations of as little as 5.6 mg/mL have been associated with 50% of maximal inhibition of LTG clearance due to VPA-mediated inhibition of UGT2B7 [126]. Interaction between VPA and LTG may lead to skin rash and even fatal Stevens–Johnson syndrome. This effect can be caused by inhibition of epoxide hydrolase enzymes and/or depletion of glutathione levels, which are involved in the detoxification of the LTG reactive metabolite [127].

CBZ and LTG block voltage-gated sodium channels and their usage in bitherapy may be associated with a higher incidence of neurotoxic adverse effects. It was reported that CBZ increased the LTG clearance by 191%. A combination of LTG with other broad-spectrum AEDs, such as LEV or TPM, might be considered as rational because both drugs had no effect on LTG clearance [125].

A decrease in LTG concentration was confirmed in the presence of rifampicin, a potent inducer of several drug-metabolizing enzymes, including CYPs and UGT [128]. Oral contraceptives, especially progestogen components, reduce LTG concentration by interacting with UGT [129]. The antiretroviral combination used in the treatment of COVID-19, including lopinavir/ritonavir, reduced serum LTG concentrations by about 50% [130], whereas atazanavir/ritonavir reduced them by about 30% [131], probably by induction of the glucuronidation enzyme system. The therapeutic efficacy of LTG may be decreased in combination with other drugs used in the COVID-19 disease, such as hydroxychloroquine or chloroquine, but the mechanism of this interaction is unknown [75].

On the other hand, LTG, as a weak inducer of UGT, may affect concentrations of antipsychotic drugs, including risperidone, quetiapine, clozapine and olanzapine [132].

Several studies on animal models proved that lower anticonvulsant activity may be observed when LTG is co-administered with aminophylline, while buprion, nebivolol, captopril, enalapril and losartan enhanced the drug action [133].

Interaction of LTG with low fat-low calorie and high fat–high calorie diets were confirmed to lead to a significant decrease in bioavailability of the drug [134].

## 10. Perampanel

Perampanel (PER), the novel drug, was approved for adjunctive treatment of partial-onset seizures and generalized tonic–clonic seizures for patients aged 12 years and older with idiopathic generalized epilepsy. It acts as a selective non-competitive α-amino-3-hydroxy-5-methyl-4-isoxazolepropionic acid (AMPA) glutamate receptor antagonist [29,37,58,96,135,136].

The pharmacokinetics of PER is linear at the dose range of 2–12 mg [48]. The steady state is reached after more than 420 h because of a long half-life (Table 1). The current reference range for PER in human plasma is 0.1–1.0 mg/L [86]. However, a higher initial target concentration of PER at 200–600 ng/mL is recommended. Serum concentrations over 600 ng/mL, however, caused greater anti-seizure effects but, at the same time, increased the risk of side effects [49].

PER, such as other newer antiepileptic drugs, has a low potential for drug interactions (Table 2). Despite its strong protein binding (in vitro), there is no significant interactions with warfarin or other drugs strongly bound to proteins. However, simultaneous administration of CYP3A4 enzymes inducers PHT, CBZ and PHB increased its metabolism and even reduced PER plasma concentration by 51%, 67% and 37%, respectively [49]. Ketoconazole inhibited PER metabolism and increaseed plasma concentrations [29,48,98,135,137].

PER increased clearance of CBZ, CLB, LTG and VPA; however, the changes were not-clinically important. In turn, OXC clearance was decreased by 26%, but clinical meaning is not defined, because an active metabolite of OXC was not determined [138].

High-fat meals delay PER absorption, but they do not affect extent absorption; then, it can be administered with or without food [48].

PER is a weak inducer of CYP2B6, CYP3A4/5 and UGTs and may decrease concentration of darunavir/cobicistat. Its activity might be impaired if co-administered with tocilizumab through hepatic enzyme induction. Ritonavir, as a potent inhibitor of CYP3A/CYP2D6, may potentially increase its level (Table 3). In patients managed with ECMO, the drug bound to proteins may be released into circulation, resulting in unpredictable effects. Therefore, clinicians should monitor serum drug concentrations to guide dosing of PER in patients [75].

## 11. Phenobarbital

Phenobarbital (PHB), 5-ethyl-5-phenylbarbituric acid), the first generation of AEDs, was introduced in 1912. It is still used worldwide for treatment of all form epilepsy, except absent seizures, as monotherapy or as adjunctive medication [19]. The World Health Organization has recommended the drug for the treatment of tonic–clonic seizures in developing countries [139]. The mechanism of action is based on the stimulation of GABAA receptors in the central nervous system (CNS), resulting in an increase in chloride ions and reduced neuronal excitability [140,141]. The elimination half-life of PHB is in the wide range of 30–173 h (Table 1). Therapeutic levels in human plasma are proposed to be in the range of 10 (15)–40 mg/L [50,51,52]. Interactions of PHB are results of the inducing effect of PHB on CYP1A2, CYP3A, 4CYP3A6, CYP2B, CYP2C9, CYP2C19 and UTGs [69,75]. Effective pharmacokinetic reactivity of PHB is noticeable, especially with other AEDs. Most interactions concern the metabolic inhibition of PHB and, finally, increased levels of PHB are observed. It is established, in the case following AEDs PHT, VPA, FBM, methsuximide, OXC, rufinamide (RFM) and STP. Moreover, PHB influences the metabolism of some AEDs and medicines in other groups. PHP increases the metabolism and decreases the level of CBZ, clonazepam, FBM, LTG, OXC, PER, PHT, RFM, zonisamide (ZNS), TGB, diazepam, clonazepam, warfarin and estrogens. Other drugs, such as dicoumarol, thioridazine, troleandomycin and strong inducer rifampicin, can decrease plasma levels of PHB [19,29]. In a study of population pharmacokinetics, PHB decreased brivaracetam (BRV) exposure by 19%, without significant effect on pharmacodynamics [142]. PHB belongs to the AEDs with potential significant interactions, including drugs used in the treatment of COVID-19. Therefore, therapeutic efficacy of PHB can be decreased in treatment with the use of hydroxychloroquine and chloroquine and tocilizumab. As a CYP2A inducer, PHB may also decrease lopinavir as well as sofosbuvir concentration. In patients with moderate or severe hepatic impairment, as well as in those with renal failure, including critically ill COVID-19 patients, a dose of PHB adjustment may be needed [75]. PHB, as an AEDs inducer, can increase the metabolism of the orally co-administered contraceptives and, finally, decrease their therapeutic effect. Moreover, among other AEDs, PHB, in comparison to LTG or LEV, possesses a high degree of malformations and neurocognitive deficits. Increased dosage of folic acid can be helpful in decreasing the risk of autism in children, whose mothers have to take AEDs. However, excess intake of folic acid can reduce the concentration of AEDs and result in an anti-seizure effect. AEDs also show interactions with food, nutrients and other vitamins. Barbiturates, including PHB, may increase the metabolism of vitamin D in the liver and, finally, may cause its deficiency [10]. After the ketogenic diet the serum concentrations of PHB slightly increased [94]. 

PHB serum concentrations were significantly increased from 7 days up to 14 days after the influenza vaccine; however, PHB is mainly metabolized by hepatic CYP2C9, but less by CYP2C19 and CYP2E1. It is known that, in case of CYP2C9, the effect of interferon-gamma is not defined [1,9].

## 12. Phenytoin and Phosphenytoin

Phenytoin (PHT) is one of the oldest, first-generation AEDs, approved in 1908 in Germany and in 1953 in the USA. Its mechanism of action is based on the modulation of voltage-gated sodium channels, enhancing their rapid inactivation. PHT and its water-soluble formulation, phospfophenytoin, are used in the prevention of tonic–clonic seizures and focal seizures [37,58,143].

PHT exists in many formulations, such as tablets, suspension and extended release and different physicochemical properties of the API and its absorption are also variable. The daily dose for adults is 300–400 mg/day in 3 doses. It can be increased up to 600 mg/day, if necessary. The PHT plasma concentration–time profile in individuals on a given dose is difficult to interpret pharmacokinetically and the drug plasma concentrations should be monitored. The reasons for that include high protein binding (Table 1) and saturable metabolism described by a non-linear model of Michaelis Menten pharmacokinetics [54]. The therapeutic range is established in the range of 10–20 mg/L and 1–2 mg/L for free fraction. It is also important to take into consideration the level of albumin. In case of decreased level of the proteins, the determined free concentration in plasma has to be normalized using the Sheiner–Tozer formula [19,29,144].

PHT, as inducer of also CYP3A4, CYP1A2 and uridine glucoronate-glucuronsyltransferase, decreases levels of CBZ, clonazepam, FBM, LTG, OXC, PER, PHB, PRM, RFM, TPM, VPA, ZNS and TGB. PHT effectively also interacts with other therapeutic classes. It decreases levels of calcium channel blockers, digoxin, quinidine, hormonal contraceptives, proton pump inhibitors, ibrutinib, nilotinib, immunosuppressant sirolimus and many statins, namely, atorvastatin, cerivastatin, lovastatin, simvastatin and warfarin [29].

There is a very significant dose-dependency of CYP3A4 induction, resulting in a 3-fold increase in CBZ clearance at high PHT doses of 20 mg/kg/day, which increases the concentration-dependent auto-induction of CBZ clearance (2.5-fold at 12 mg/L) [95]. The clearance of busulfan is altered as result of the interaction with PHT and changes in sub or supra concentrations can cause an increase in toxicity [145].

During the SARS CoV-2 pandemic, attention is paid to the interaction of PHT with antiretrovirals used to treat COVID-19. The use of PHT together with lopinavir/ritonavir combination can reduce the antiretrovirals by 30%. In that situation, COVID-19 patients with epilepsy may require an increase in the lopinavir/ritonavir dose of about 50% to maintain the therapeutic range [8,146]. PHT should be used in caution with nitazoxanide because its metabolite tizoxanide is highly bound to plasma proteins (>99.9%) and may displace PHT, leading to an increase in its pharmacological effect [75]. Earlier data stated that serum PHT levels may increase temporarily in some people after influenza vaccination. Transient increases in concentration by 46–170% have been confirmed [9,147].

As other AEDs, PHT influences the increase in the metabolism of vitamin D in the liver and may cause deficiency of vitamin D; hypocalcemia is observed to increase homocysteine concentrations in patients treated with PHT. Moreover, excess supplementation of folic acid decreases the level of PHT and therapeutic effectiveness of the drug. PHT may also cause deficiency of vitamin K. Body weight is not affected by PHT treatment [10].

Fosphenytoin (FOS) is a prodrug of PHT used in the treatment of critically ill patients with focal-onset and generalized-onset tonic–clonic seizures, if oral administration is impossible. FOS is rapidly absorbed after intramuscular (i.m.) administration. Metabolism and renal excretion are the same as PHT and the drug represents nonlinear pharmacokinetics (Table 1). Drug–drug interactions are the same as PHT (Table 2 and Table 3).

## 13. Piracetam and Its Newer Derivatives

Piracetam (PIR) is a derivative of GABA but its pharmacological action results from the restoration of cell membrane fluidity. PIR improves cerebral microcirculation due to the reduction of erythrocyte adhesion to the vascular endothelium and an influence on blood coagulation and blood vessels. The drug has neuroprotective activity and modulates the cholinergic, serotonergic, noradrenergic and glutamatergic neurotransmission. It is effective in cognitive disorders and dementia, vertigo, cortical myoclonus, dyslexia and sickle cell anemia. The pharmacokinetics of PIR is linear with complete bioavailability in the oral form, no protein binding and excretion of the unchanged form with no metabolites (Table 1). Therefore, PIR has no profound pharmacokinetic interactions with other medications [57]. However, there are several pharmacodynamic interactions. The combinations of PIR with clonazepam or VPA are favorable in patients with progressive myoclonus epilepsy due to synergistic action of the drugs [148]. It was shown that the drug and its analogues act as competitive antagonists of barbiturate, diazepam and melatonin inhibition of glucose transport in the brain [149]. Due to the PIR influence on coagulation and platelet function, the drug may have potential interactions with antiplatelet drugs such as clopidogrel [150].

The anticonvulsant activity of LEV and BRV is connected with binding to the synaptic vesicle protein 2A (SV2A), which is implicated in synaptic signal transduction. In addition, LEV inhibits N-type channels and α-amino-3-hydroxy-5-methyl-4-isoxazolepropionic acid (AMPA) glutamate receptors responsible for the majority of fast excitatory transmission in the CNS. LEV is indicated for the treatment of partial-onset seizures, myoclonic seizures and generalized tonic–clonic seizures, while the recommendation for BRV includes partial-onset seizures in children 4 years of age and older. The pharmacokinetics of LEV is linear, with rapid and almost complete absorption and distribution into the cerebrospinal fluid compartment (Table 1). LEV is neither bound to plasma proteins nor metabolized in the liver by the CYP450 system [20]. There is one inactive metabolite LO57 which is formed by hydrolysis of the acetamide group [46]. Therefore, drug–drug interactions are not common and LEV is a suitable candidate for combination therapies. Co-administration of LEV with other AEDs that enhance GABAergic inhibition may be considered for rational polytherapy [151]. Synergistic interactions were reported among LEV and CBZ, PHT, TPM and VGB [152]. In the mouse maximal electroshock-induced seizure model, FBM increased LEV concentration in the brain, leading to an additive anticonvulsant effect [153]. It was shown that perindopril arginine may positively influence the pharmacological action of LEV in epileptic patients [154].

LEV does not interact with food. However, patients with epilepsy receiving LEV had higher zinc, selenium, copper, iron, aluminium, cadmium, cobalt and nickel levels than healthy controls. It was suggested that the increase in the trace elements levels may be caused by their increased absorption from the gastrointestinal (GI) tract or decreased elimination from body in patients treated with LEV [155].

BRV exhibits low protein binding and extensive metabolism [20,21]. A recent review suggested that the drug did not cause any clinically significant interactions by impairing drug-metabolizing CYP enzymes or drug transporters [156]. However, several interactions with AEDs and other drugs were reported, which influenced pharmacokinetic parameters. As a moderate reversible inhibitor of epoxide hydrolase, BRV can affect the hydrolysis of CBZ and its neurotoxic epoxide into inactive metabolites [157]. This effect was more pronounced in presence of VPA [158]. Coadministration of BRV with PHT led to increasing in PHT C_max_ and AUC, due to the inhibition of CYP2C19 engaged in PHT elimination [157]. BRV co-administered with combined oral contraceptives (ethinylestradiol, 30 μg, and levonorgestrel, 150 μg) decreased the AUC of the hormone by 27% and 23%, respectively [159]. On the other hand, inducers and inhibitors of CYP enzymes may affect BRV pharmacokinetics. Patients receiving adjunctive treatment with CBD showed increases in BRV levels by 95–280%, as a result of the inhibition of CYP2C19 by CBD [160]. CBZ, PHT and PHB/PRM decreased BRV levels in epileptic patients by 26%, 21% and 19%, respectively. However, influence on clinical response was not significant [142]. Similar results were obtained in healthy volunteers, who exhibited reduced BRV AUC by 29% after CBZ coadministration [161]. The above changes were caused by increased BRV clearance due to CBZ-induced CYP activity. A potent inducer of CYP3A4, 2B6, 2C8, 2C9 and 2C19 antibacterial agent rifampin decreased BRV exposure by 45% without a change in C_max_ in adult epileptic patients [162].

BRV and LEV could be safe treatment options for patients with COVID-19 suffering from severe respiratory and/or cardiac problems, because these drugs are devoid of respiratory/cardiac adverse effects. Lopinavir/ritonavir decreases the plasma concentration of BRV, while the therapeutic efficacy of the AED may be increased when used in combination with chloroquine, hydroxychloroquine and ribavirin (Table 3). The BRV dose adjustment is recommended for patients with all stages of hepatic impairment. No significant drug interactions were reported for LEV and anti-COVID-19 medications. In patients with renal impairment dosage adjustment is required [75].

## 14. Primidone

Primidone (PRM) was approved in 1954 to treat partial-onset and generalized seizures, as well as essential tremors. The mechanism of action involves synaptic and extrasynaptic binding to the GABA receptor. PRM represents linear pharmacokinetics. PRM undergoes extensive hepatic metabolism to PHB and PEMA—phenylethylmalonamide (Table 1). The final metabolite is responsible for the main pharmacodynamics action of PRM. The reference range for PRM in monotherapy is 5–10 mg/L, but its active metabolite PHB should also be monitored during the treatment with PRM. PRM has significant potential to interact like PHB as a result of its inducing effect on CYP1A2, CYP3A6, CYP2B, CYP2C, CYP3A4 and UTGs [19,29,58,59].

The metabolism of PRM can undergo two directions, induction as well as inhibition. PRM decreased LCM plasma levels by 30–40% [163].

PRM also decreases levels of other AEDs, such as CBZ, clonazepam, FBM, LTG, OXC, PER, PHT, RFM, ZNS and TGB. It interacts with other therapeutic classes, causing a decrease in levels of beta-blockers, calcium channel blockers, digoxin, hormonal contraceptives and statins, namely, lovastatin, simvastatin, cerivastatin, atorvastatin and vitamin D [29]. Moreover, some AEDs can decrease the plasma level of PRM. Among the drugs are CBZ and PHT. Moreover, CLB, ESM and STP increase PRM concentrations. An interesting situation is observed in the case of sulthiame (STM) and VPA, which were not affected by PRM, but they increased the plasma level of PHB—its active metabolite [29].

The latest study, an in vitro and in murine model, provided proof that PRM can be a potential drug to inhibit the receptor-interacting serine/threonine protein kinase 1, which is a key mediator of regulated cell death and inflammation. Therefore, PRM has the potential to be an anti-inflammatory agent in diseases with hyperinflammation, such as COVID-19 [164]. Similarly to PHB, in patients with moderate or severe hepatic impairment, as well as in those with renal failure, including critically ill COVID-19 patients, correction of the dose of PRM may be needed [75].

PRM, such as PHB or PHT, decrease exposure to the drugs used in the treatment of COVID-19, such as atazanavir, darunavir/cobicistat, lopinavir/ritonavir, remdesivir, chloroquine and hydrochloroqiune and should not be co-administered with these anti COVID-19 medications [75].

PRM may increase the metabolism of vitamin D in the liver and, finally, may cause its deficiency [10].

## 15. Rufinamide

Rufinamide (RFM) is a newer anticonvulsant approved for the treatment of the Lennox–Gastaut syndrome. The mechanism of action of RFM involves limitation of sodium-dependent potentials, resulting in a membrane stabilizing effect. The drug shows non-linear pharmacokinetics and the absorption is not directly proportional to the dosage due to limited dissolution [60]. The bioavailability after oral administration is enhanced by food (Table 1). RFM has low protein binding and undergoes carboxylesterase-mediated enzymatic hydrolysis to inactive carboxylic acid derivative (CGP 47292) [61]. Plasma RFM concentrations can be increased by co-administration of VPA due to inhibition of the carboxylesterases [60]. It was reported that CBD increased serum concentrations of RFM and this effect was dose-dependent (*p* = 0.004). However, the mechanism of the interaction is unknown [23]. On the other hand, CBZ, VGB, PHT, PHB or PRM can decrease steady-state plasma RFM concentrations [60]. The drug is a weak inducer of CYP3A4 enzymes. When co-administered with CBZ, it increased its clearance in children and adults, leading to a reduction in steady-state plasma CBZ concentrations. A similar effect was observed for a combination of RFM and LTG, possibly due to the induction of uridine diphosphate glucuronosyltransferase (UDP-GT), responsible for LTG elimination [112]. Interactions with other drugs involve triazolam and oral contraceptives [60]. RFM decreased the exposure to ethinyl oestradiol and norethindrone as a result of the induction of CYP3A4 and/or UDP-GT [165]. The same mechanism may be associated with the interaction with antiviral drugs used in the treatment of the COVID-19 patients, as RFM decreases exposure to atazavir, darunavir/cobicistat, lopinavir/ritonavir, remdesevir, chloroquine and hydroxychloroquine (Table 3).

Interaction with food exerts a positive effect on the exposure to RFM, as food facilitates its absorption in the bowel. High-fat meal increased C_max_ and AUC of the drug by 36–96% and 31–44%, respectively, depending on the RFM dose and formulation [60].

## 16. Stiripentol

Stiripentol (STP) is indicated for use in conjunction with CLB and VPA as adjunctive therapy of refractory generalized tonic–clonic seizures in patients with Dravet syndrome [166] and can be effective in other rare epilepsy syndromes and refractory status epilepticus [167].

The drug activates GABA_A_ receptors and inhibits lactate dehydrogenase [167]. It exhibits dose-dependent pharmacokinetics showing reduced clearance and change in elimination half-life (Table 1) from 2 to 10 h with increased dose [62]. The drug possesses low bioavailability, binds highly to plasma proteins and exhibits extensive hepatic metabolism (Table 1). It inhibits CYP450 enzymes, thus increasing the plasma concentrations of many other antiepileptic drugs [63]. The most prominent interaction is with CLB, which is used in combination with STP in Dravet syndrome. STP inhibits CYP3A4 and CYP2C19 engaged in the demethylation of the CLB active metabolite, leading to a two-fold increase in the concentration of the parent drug and from a three- to five-fold increase in concentration of its metabolite [167]. Plasma concentrations of other AEDs, including CBZ, PHB, PHT, VPA, diazepam, ESM and TGB, may be increased with STP co-administration [168,169,170,171]. Concomitant administration of CBD and STP may increase STP C_max_ 1.3-fold and AUC 1.6-fold, while 7-OH-CBD and 7-COOH-CBD exposure were decreased by 29% and 13%, respectively [172].

STP has interactions with other drugs metabolized by CYP3A4, such as antihistamines, antiemetics, anesthetics, analgesics, antibiotics, cardiac agents and immune modulators and agents used in neuropsychiatry and oncology. It increases the concentration of drugs metabolized by CYP2C19, including certain antidepressants, proton pump inhibitors, clopidogrel and cyclophosphamide [167]. Currently, there is no evidence for any significant drug interactions between STP and medications used to treat COVID-19 patients.

## 17. Sulthiame

Sulthiame (STM), a cyclic sulfonamide, was synthesized for the first time in the 1950s by Bayer and is less commonly used now. The mechanism of the pharmacological activity of STM involves the inhibition of carbonic anhydrase. The drug was used for the treatment of benign epilepsy with centrotemporal spikes in 4–12-year-old children [173]. Children receiving a comparable dose of STM per body weight have lower levels of STM than adults [174].

The most recent data point to a non-linear STM pharmacokinetics in healthy volunteers. Data calculated using non-compartmental pharmacokinetic analysis showed that plasma clearance decreased several folds with increasing dose (after 50, 100 and 200 mg and amounted to 40.4, 11.7 and 8.3 L/h) (Table 1). Bioavailability after oral administration was complete and the protein binding was low [65].

CBZ, PRM and PHT increase the elimination of STM. Antacids with magnesium oxide or magnesium trisilicate and bismuth oxycarbonate decrease GI absorption of STM and, in consequence, decrease plasma levels of STM [19,65]. Serum PHT levels have been shown to increase when STM is added to PHT therapy. STM can also lead to an increase in PHB levels in the blood. The therapeutic reference range of STM newly established is 2–10 mg/L (5–35 µmol/L) [86].

A study on the effect of STM on the pharmacokinetics of CLB confirmed the interaction between STM and CLB. As a result, STM inhibited the metabolism of CLB in adults and children. The mechanism of the interaction is, most probably, the inhibition of CYP2C19 enzyme activity [88].

The use of atazavir, darunavir/cobicistat and lopinavir/ritonavir can potentially increase exposure to STM (Table 3).

## 18. Topiramate

Topiramate (TPM) is a second-generation AED approved for the treatment of focal and primary generalized-onset tonic–clonic seizures, including Lennox–Gastaut syndrome. The drug is also applied in bipolar disorder, post-traumatic stress, mood instability disorder, bulimia nervosa, weight loss in obese and to prevent migraine headaches. Mechanisms of action involve inhibition of voltage-dependent sodium and calcium channels, decreasing carbonic anhydrase activity and enhancing the inhibitory effect of GABA [58].

TPM absorption was linear over a wide dose range, after a single-dose in healthy volunteers in the range of 100–1200 mg TPM. Both C_max_ and AUC were linear and increased in proportion to dosage. TPM binded poorly to plasma proteins (Table 1). Elimination of TPM was mainly renal, with renal clearance of 0.6–1.2 L/h; therefore, TDM is especially necessary in patients with renal impairment. In the absence of hepatic enzyme induction, 80% of the TPM dose was excreted unchanged in the urine of healthy volunteers. TPM does not appear to have the ability to alter the metabolism of most drugs, but is itself the target of drugs known as liver enzyme inducers. CBZ, PHT and VPA reduced TPM plasma concentrations by 48% 40% and 17%, respectively [69,70,71]. More recent data confirmed TPM interactions. TPM clearance increased with the combined use of PHT, CBZ, OXC and PHB. The increase was up to 2.52 L/h after PHT, 2.17 L/h after CBZ, 1.803 L/h after OXC and 1.636 L/h after co-administration of PHB and was 117, 87, 55 and 41% higher, respectively, than in patients treated with TPM without concomitant treatment [175].

In monotherapy, 20–30% of TPM is metabolized, while, when administered together with CBZ and PHT, the metabolism increases up to 50–70%. The increase in elimination also causes ESL. The TPM biological half-life becomes up to 50% shorter after administration with enzyme-inducing AEDs. Over 200 drugs are known to interact with TPM [100,175]. TPM clearance is also decreased by lithium, propranolol, amitryptiline and sumatriptan, diltiazem, hydrochlorthiazide, metformin, propranolol and, finally, posaconazole, increasing the level of TPM in serum [98]. Moreover, TPM has a dose-dependent interaction with contraceptives containing estrogen [1]. A new therapeutic reference range of 2–10 mg/L (5–35 µmol/L) was proposed for TPM, which is smaller than the previous 5–20 mg/L [86].

TPM is CYP3A4 inducer and may interact with anti-COVID-19 medication increasing darunavir/cobicistat and lopinavir/ritonavir concentrations (Table 3). In patients with COVID-19 and liver or renal disfunctions TPM dosage may need to be adjusted [75].

CBD interacts with TPM in humans. Increasing doses of CBD from 5 mg/kg to 50 mg/kg/day increased significantly TPM concentration in serum, which emphasizes the need for therapeutic monitoring of TPM [23,76].

Food delayed oral absorption by about 2 h, but does not affect TPM concentrations. TPM has an impact on body weight [10]. After the dietary therapies, the serum concentrations of TPM decreased, but not significantly [94].

## 19. Valproic Acid

Valproic acid (VPA) is used as an anticonvulsant and mood-stabilizing drug. The drug increases the activity of GABA through inhibition of GABA degradation, inhibition of GABA transaminobutyrate, increased GABA synthesis and decreased turnover. Moreover, VPA blocks voltage-gated Na^+^, Ca^2+^ and K^+^ channels and also acts as a histone deacetylase (HDAC) inhibitor [176] The latter mechanism is responsible for the VPA anticancer potential to alter the expression of many genes, including tumor suppressor genes [177].

VPA is highly protein bound (Table 1) and extensively metabolized in the liver. Concomitant use of cytochrome CYP inducers or inhibitors can result in altered VPA absorption or elimination rates [72]. STP significantly increases serum VPA concentrations by inhibiting the CYP2C19 pathway, especially when administered concomitantly with TPM [178]. ESM can decrease the drug levels in children with epilepsy [179]. VPA concentrations may be elevated when co-administered with FBM, due to an inhibition of β-oxidation [180].

Moreover, carbapenems can reduce urinary elimination of VPA by irreversible inhibition of acylpeptide hydrolase, an enzyme responsible for the deconjugation of valproate–glucuronide complex [181,182]. VPA levels and efficacy may decrease when it is co-administered with protease inhibitors, such as ritonavir, due to ritonavir-mediated induction of VPA glucuronidation [183].

A reduction in VPA concentration by 50% was observed during cisplatin-based chemotherapy due to impaired intestinal absorption [184].

VPA acts as a weak CYP inhibitor and increases the concentration of LTG [125], PHB [185] and RFM [60]. When co-administered with CBZ, VPA may inhibit epoxide hydrolase and causes an increase in the concentration of the active epoxide metabolite CBZ-E, leading to CNS side effects. However, a newer study suggests that toxicity after combination therapy of VPA with CBZ may be caused by the inhibition of the conversion of CBZ-E to trans diol metabolite and inhibition of further glucuronidation [186].

By inhibiting various CYP isoenzymes, VPA increases the toxicity of such chemotherapeutic drugs such as nitrosoureas, cisplatin and etoposide [187]. On the other hand, the drug induces CYP3A4 and P-glycoprotein engaged in the pharmacokinetics of rivaroxaban, leading to a decrease in plasma levels of the anticoagulant and recurrence of deep-vein thrombosis [188].

Pharmacodynamic interactions can be observed between VPA and AEDs or other drugs. Potentially favorable pharmacodynamic interactions are reported between VPA and LTG, VPA and ESM in patients with absence seizures and between VPA and CBZ in patients with partial seizures [189]. Synergistic effects were observed when VPA was co-administered with antipsychotics in patients with mania [190]. A combination of VPA and TPM may enhance the risk of VPA-associated side effects, such as hyperammonemia, thrombocytopenia and hypothermia [191]. In patients taking concomitant VPA and CBD elevated liver function test results (ALT, AST) were noted [23].

Potential interaction of VPA with drugs used in COVID-19 disease can be predicted for remdesivir and lopinavir, leading to changed exposure to antiviral drugs [76].

Several studies have shown that VPA is influenced by food intake. A diet containing soybeans may cause lower plasma concentrations of VPA and elevated levels of GABA in the brain, due to UGT induction [192]. A high-fat diet during VPA treatment leads to increased hepatotoxicity via modulation of mitochondrial β-oxidation [193]. Liver dysfunction was also observed during the combined anticonvulsant therapy involving VPA and the ketogenic diet [194]. It was reported that long-term therapy with VPA causes a decrease in vitamin D levels in children by inducing enzymes that are responsible for the catabolism of vitamin D [195]. The deficiency of the vitamins from the B group, including folic acid, was also observed [196]. The drug inhibits the glutamate formyltransferase enzyme and decreases the formation of the active metabolite of folic acid [197]. It was suggested that VPA combined with vitamin B12 or vitamin B complex can improve epilepsy control after stroke [198]. Moreover, Karabiber et al., suggested that VPA significantly increases homocysteine levels in epileptic patients, which may lead to poor seizure control and the development of refractory epilepsy [196].

Caution is warranted when lopinavir/ritonavir and darunavir/cobicistat are administered with VPA, as there is a potential for altered concentration of the antiviral drugs (Table 3). Moreover, lopinavir/ritonavir decreases the plasma concentrations of VPA, probably by induction of the glucuronidation enzyme system [183]. In critically ill patients with COVID-19, abnormal liver function may happen. In patients with moderate or severe hepatic impairment, VPA may need dosage adjustment [75].

## 20. Zonisamide

Zonisamide (ZNS) is a benzisoxazole derivative used as an adjunct treatment for partial seizure epilepsy and label for bipolar disorder, chronic pain, migraine and myoclonic dystonia. The drug exerts its pharmacological effects by blockage of neuronal voltage-dependent sodium and T-type calcium channels, thereby suppressing neuronal hyper-synchronization. Moreover, ZNS modulates GABAergic and glutamatergic neurotransmission and is a weak inhibitor of carbonic anhydrase.

ZNS is rapidly absorbed after oral administration. It preferentially accumulates in red blood cells. In recommended doses, it follows linear pharmacokinetics with metabolism via phase I and phase II biotransformation pathways (Table 1).

Pharmacokinetic interactions are rare and of little clinical relevance [199]. CBZ, PHT, PRM and risperidone enhance the elimination of ZNS and decrease its plasma concentration [200]. The current therapeutic range for ZNS in plasma is 10–38 (40) mg/L [29,86,87].

During pregnancy, ZNS serum concentrations may decrease over 40%, with large interindividual variability [201].

Coadministration of ZNS and potent inhibitors of CYP3A4, darunavir/cobicistat and lopinavir/ritonavir compositions could potentially increase concentrations of ZNS, which is partly metabolized by CYP3A4 [76] (Table 3). COVID-19 patients with renal failure may need ZNS dosage adjustment [75].

The interaction of ZNS with CBD was also observed. The increase in CBD doses caused increased serum concentrations of ZNS in adult patients [23].

## 21. Conclusions

Polytherapy may be a rational strategy in the treatment of many patients and clinicians should be familiar with the main pharmacokinetic characteristics of AEDs and their potential for interactions. One has to keep in mind that epileptic seizures occur during the progression of many diseases, including HIV infections or cerebrovascular disease. Moreover, AEDs are widely used in the long-term treatment not only of epilepsia but also of neuropathic pain or psychiatric disorders. The older AEDs cause interactions involving enzyme induction or inhibition, affecting efficacy of other AEDs, but also anticoagulants, immunosuppressants, or macrolides. Newer AEDs, including cenobamate, levetiracetam, perampanel, or pregabalin, have a less potential for pharmacokinetic interactions than older AEDs and can therefore be recommended for polytherapy in epileptic patients and particular groups, such as patients with cancer, after transplantations, with anticoagulant treatments or HIV infection. CBD, which is increasingly used in the therapy of certain types of epilepsy, exhibits interactions with CBZ, ESL, BRV, RFM, STP and TPM.

During the pharmacotherapy of COVID-19 disease, concomitant use of the first-generation AEDs with antiviral drugs such as atazanavir, remdesivir and drug compositions (darunavir/cobicistat, lopinavir/ritonavir) should be evaluated carefully in terms of drug interactions leading to toxicity or treatment failure. The use of CBZ, PHT, PHB and PRM potentially decreases exposure to anti-COVID drugs. On the other hand, the use of atazavir and darunavir/cobicistat can potentially increase exposure to CBZ. Besides, the use of darunavir/cobicistat can potentially decrease exposure to PHB. Currently, there is no evidence for any significant drug interactions among favipiravir, interferon beta and oseltamivir with AEDs. Cardiac, hepatic, or renal impairments in patients with COVID-19 may require ECG/QT interval monitoring and/or adjustment of AEDs dosage. Moreover, ECMO, which may become necessary for patients with COVID-19 and severe pneumonia, may affect plasma concentration of highly protein-bound drugs, including clobazam, perampanel, phenytoin and valproate, leading to increase in their pharmacological action.

It was proved that specific diets are effective in the management of certain types of epilepsy. Generally, taking AEDs with food may cause prolonged drug absorption but without effect on plasma concentrations. However, in the case of RFM co-administration with food, this leads to a significant increase in bioavailability, while the opposite effect was observed for LTG. On the other hand, AEDs may influence the nutritional health of patients leading to deficiency of vitamins (K, D and B group), increased level of homocysteine or changes in levels of trace elements. Therefore, monitoring of dietary intake, serum levels of nutrients and compensations of deficiencies is recommended in epileptic patients for general health and better seizure control.

## Figures and Tables

**Table 1 ijms-22-09582-t001:** Pharmacokinetic parameters of AEDs.

Drug	Absorption (%)	tmax (h)	Bioavailability (%)	Protein Binding (%)	Vd/F(L/kg)	Metabolic Pathway	Metabolites	Excretion	Cl/F	t0.5 (h)	Ref.
Brivaracetam (BRV)	~100	0.5–2	~100	≤20	0.5	extensive metabolized by hydrolysis via amidase enzymes and oxidation by CYP2C9 and CYP2C19	3 inactive metabolites (acid, hydroxy and hydroxyacid)	>95% excreted in urine (10% unchanged)	0.7–1.07 mL/min/kg	7–8	[19,20,21,22]
Cannabidiol (CBD)	low	0.5–4	5–19 oral 13–31 inhaled	94–99	19.5–32.7	metabolized in the liver by cytochrome P450 isoenzymes, including CYP2C9, CYP2C19, CYP2D6 and CYP3A4, to the active metabolite 7-hydroxy-CBD and further to inactive metabolites	active metabolite 7-hydroxy-cannabidiol (7-OH-CBD)	excreted in urine and feces	2546–4741 L/h in fasted stated after multiple dose p.o., 533 L/h in fed state p.o., 3252–3783 L/h single doses p.o., 74.4 L/h i.v. injection	1.4–10.9 p.o. spray, 1.1–2.4 nebulizer, 1.0–3.2 oral, 24 i.v. infusion, 31 smoking, 2–5 days chronic oral	[18,22,23,24,25,26,27,28]
Carbamazepine (CBZ)	85	19 ± 7 after a single dose 5.9 ± 1.8 after multiple dosing	75–85	75–80	0.8–2.0	extensively (>90%) metabolized in the liver by CYP3A4 and CYP2C8	main active metabolite carbamazepine-10,11-epoxide (CBZE)	mainly excreted as hydroxylated and conjugated metabolites and <1% unchanged	25 ± 5 mL/min after a single dose and 80 ± 30 mL/min after multiple dosing	18–65 after a single dose, 10–20 after multiple dosing	[19,22,29,30]
Cenobamate (CNB)	88	1–4	88	60	0.6–0.7	extensively metabolized by glucuronidation via UGT2B7 and oxidation via CYP2E1, CYP2A6, CYP2B6 and less by CYP2C19 and CYP3A4/5 isoenzymes	eight inactive metabolites	unchanged drug and its metabolites are excreted with urine (88%) and feces (5%)	0.5–1.4 L/h	50–60	[22,31,32,33,34]
Clobazam (CLB)	>87	1–3	≥95	80–90 CBZ70 N-CLB	0.9–1.8	extensive metabolized in the liver by CYP3A4 isoenzyme (98%) to an active metabolite N-desmethylclobazam (N-CLB) and other metabolites	main active metabolite N-desmethylclobazam (N-CLB) and over 20 other metabolites	eliminated via urine (~94%) as metabolites	1.9–2.3 L/h	10–30 CLB, 36–46 N-CLB	[19,22,29,35]
Eslicarbazepine Acetate (ESL)	>90	1–4	>90	<40	0.88	extensively metabolized to eslicarbazepine (95%) by hydrolytic first-pass metabolism	main active metabolite S-licarbazepine	eliminated primarily via renal excretion	3 ± 0.7 L/h	9–20	[19,22,29,36]
Ethosuximide (ESM)	90–100	1–4	>90	0–22	0.6–0.7	hepatic extensive metabolized mainly by CYP3A4 and CYP2E1 isoenzymes	no active metabolite	excretion mainly via the kidneys and biliary, 10–20% excreted unchanged	0.01 ± 0.004 L/h/kg	40–60 adults, 30–40 children	[19,29,37,38,39,40]
Felbamate (FBM)	>90	2–6 h	>90	20–48	0.7–1.0	50% metabolized in the liver by the CYP3A4 and CYP2E1 to p-hydroxy and 2-hydroxy FBM	active metabolite atropaldehyde and multiple inactive metabolites	around 50% is excreted unchanged in urine	26 ± 3 mL/h/kg (single dose), 30 ± 8 mL/h/kg (multiple dosing)	16–25	[19,22,29]
Gabapentin (GBP)	Saturable, maximum of 5 g/day can be absorbed	2–4	27–80	<3	0.8	undergoes little or no metabolism	no active metabolite	excreted unchanged by the kidney	~125 mL/min	5–7	[22,41,42]
Lacosamide (LCM)	~100	1–4	~100	<15	0.6–0.7	metabolized in the liver by demethylation, primarily byCYP2C19 to inactive O-desmethyl derivatives	inactive O-desmethyl- lacosamide derivatives	mainly renal excretion (40% unchanged, 30% as O-desmethyl-lacosamide and 20% polar fraction),<0.5% in faeces	2.13 L/h	12–14	[19,29,43,44]
Lamotrigine (LTG)	≥95	1–3	>95	55–66	0.9–1.3	extensively metabolized in the liver via UGT1A4, 1A1 and 2B7 isoenzymes to form 2-N and 5-N glucuronides	inactive metabolites 2-N and 5-N glucuronide	renal excretion (<10% unchanged)	0.18–1.21 mL/min/kg	15–35	[19,22,29,45]
Levetiracetam (LEV)	≥95	1.3–5.2	>95	<10	0.5–0.7	not metabolized in the liver by CYP450 system	inactive carboxylic acid metabolite LO57	66–70% excreted unchanged in urine and 24–27% as inactive metabolites	0.96 mL/min/kg	6–8	[19,20,22,29,46]
Oxcarbazepine (OXC)	>90	1–5	>95	60 40 (MHD)	0.7–0.75	extensively metabolized in the liver to MHD via reduction, then by glucuronidation and hydroxylation via CYP isoenzymes	main active metabolite 10-hydroxy carbazepine (MHD)	primarily excreted with urine (>95%), less than 1% unchanged, <4% fecal excretion	84.9 L/h	1–5 7–20 (MHD)	[19,22,29,47]
Perampanel (PER)	100	0.5–2.5	~100	>95	1.1	extensively metabolized in the liver, primarily via CYP3A4 and CYP3A5, as well as CYP1A2, CYP2B6, followed by glucuronidation	no active metabolitesand various inactive metabolites	eliminated in feces and urine	~13 mL/min adolescents,~10–11 mL/min elderly, adults	~105	[19,22,29,48,49]
Phenobarbital (PHB)	70–90	2–4	>90	48–55	0.5–1.0	extensively (>70%) metabolized in the liver, mainly by isoenzymes of cytochrome CYP2C9 and less by CYP2C19 and CYP2E1	main metabolite p-hydroxy phenobarbital	excreted primarily by the kidneys, 20–40% unchanged	0.06–0.23 L/kg/day	53–118 adults, 400 newborns	[19,22,29,50,51,52,53]
Phenytoin (PHT)	90–100	7–42	70–100	>90	0.5–1.0	hepatic metabolism in 98% by CYP2C9 and CYP2C19 isoenzymes	no active metabolites	excreted as inactive metabolites in the bile, <5% unchanged in urine	0.0174 L/h/kg	7–42	[19,22,29,54,55]
Phosphenytoin (FOS)	100	~0.5	~100	95–99	0.04–0.13	metabolized by phosphatases, to phenytoin, phosphate and formaldehyde	phenytoin (PHT)	renal excretion is the same as PHT	12.9–22.8 L/h	15 min (FOS to PHT)	[19,22,29,56]
Piracetam (PIR)	~100	0.5–1.5	~100	not protein bound	0.6	neither metabolized by the liver	no active metabolites	~90% is excreted in the urine as unchanged drug	80–90 mL/min	4–6	[19,22,57]
Pregabalin (PGB)	≥90	1–2	>90	not protein bound	0.5–0.6	undergo little or no metabolism (<2%)	N-methylated derivative of PGB	excreted virtually unchanged by the kidneys	~70 mL/min	5–7	[19,22,41]
Primidone (PRM)	>60	2–4	>90	20–45	0.5–0.8	extensive hepatic metabolism by CYP2C9 and CYP2C19 isoenzymes	active metabolites-phenylethylmalonamide (PEMA) and PHB	excreted in urine (40–50% unchanged)	30 mL/min	7–22 adult, 5–11 child, 8–80 newborn	[19,22,29,58,59]
Rufinamide (RFM)	≥85	4–6	70–85	26–35	0.7–1.2	extensively metabolized by carboxyesterases into inactive carboxylic acid derivative CGP 47,292 via hydrolysis	no active metabolites	>90% renally excreted (66% as CGP 47292, 2% as unchanged drug) and about 9% in feces	3.0–3.5 L/h children, 4.9–5.6 L/h adolescents and adults	6–10	[19,22,29,60,61]
Stiripentol (STP)	>70	0.5–2	low	96–99	1.0	extensively metabolized in the liver, primarily by demethylation, glucuronidation and metabolism involving the enzymatic activity of CYP1A2, CYP2C19 and CYP3A4 isoenzymes	13 metabolites	>73% renal excretion	8–40 L/kg/day	2–13	[19,22,62,63,64]
Sulthiame (STM)	100	1–5	100	29	64.8	moderate metabolism in the liver by unknown isoenzymes	unknown metabolites	renal elimination (80–90%)	8–40 L/h	5–7	[19,65,66]
Tiagabine (TGB)	≥90	0.5–2	≤90	>95	1.0	extensively metabolized by CYP3A4	at least 4 pharmacologically inactive metabolites	25% excreted in urine and 63% in feces, primarily as metabolites, and 2% unchanged	109 mL/min	5–9	[19,22,29,67,68]
Topiramate (TPM)	≥80	2–4	>80	9–20	0.6–0.8	metabolized mainly by glucuronidation, hydroxylation and hydrolysis and sulfonation, 50% undergoes hepatic metabolism by CYP isoenzymes	no active metabolites	mainly excreted in urine, 70–80% unchanged	22–36 mL/min	20–30	[19,22,29,69,70,71]
Valproic acid (VPA)	100	1–7	~90	74–93	0.1–0.2	extensively metabolized in the liver by glucuronidation (50%), β-oxidation (40%) and oxidation by cytochrome P450 (10%)	major metabolite: valproate glucuronide	<3% is excreted unchanged in the urine	0.56 L/h/m^2^	6–17	[19,22,29,72]
Vigabatrin (VGB)	60–80	1–2	60–80	not protein bound	0.8–1.0	undergoes little or no metabolism by the liver	no active metabolites	95% eliminated in urine, of which ~80% is unchanged	2.4 L/h infants, 5.1 L/h children, 5.8 L/h adolescents,10.5 L/h adults	5–8	[19,22,29,73]
Zonisamide (ZNS)	≥90	2–5	>90	40	1.0–1.9	extensively metabolized in the liver, primarily by acetylation, to form N-acetyl zonisamide and reduction by CYP3A4 to form 2-sulfamoylacetylphenol	no active metabolites	35% excreted primarily in urine as unchanged form and as the glucuronide of a metabolite	0.30–0.35 mL/min/kg	27–70	[19,22,29]

**Table 2 ijms-22-09582-t002:** Changes in plasma concentrations expected when an antiepileptic drug (AED) is added to a pre-existing AED regimen.

Added AED			Pre-Existing AED																								
		BRV^h^	CBD^h^	CBZ^c,h^	CLB	CNB^c,h^	ESL-a^r^	ESM	FBM	GBP^r^	LCM^c,h^	LTG^h^	LEV^r^	OXC	PER^h,r^	PHB^c,h^	PHT^c,h^	PGB^c,r^	PRM^r^	RFN^c^	STM	STP	TGB	TPM^h,r^	VPA^h^	VGB	ZNS^r^
BRV			BRV ▲	CBZ ▲ CBZ-E ▲													PHT ▲										
CBD					CLB ▼ DMCLB ▲		ESL ▲												PRM ↓	RFN ▲		STP ↑		TPM ▲			ZNS ▲
CBZ			CBD ↓	BRV ↓	N-CLB ↑		ESL ↓	ESM ↓	FBM ▼		CBZ ↓	LTG ▼		H-OXC ↓					PRM ↑	RFN ↓	STM ↓		TGB ▼				ZNS ▼
CLB																											
CNB				CBZ ▼							LCM ↓	LTG ▼	LEV ↓			PHB ▲	PHT ▲										
ESL-a																	PHT ↑								VPA ↓		
FBM				CBZ ▼CBZ-E ▲												PHB ▲	PHT ▲								VPA ▲		
ESM																			PRM↑								
GBP									FBM ↑								PHT ↑										
LCM				LCM ↓													PHT ↑		PRM ↓					LCM ↑			
LTG																PHB ↓									VPA ↓		
LEV																											
OXC												LTG ↓				PHB ↑	PHT ↑			RFN ↓				TPM ↓			
PER												LTG ↓															
PHB			CBD ↓	CBZ ↓	N-CLB ↑		ESL ↓	ESM ↓	FBM↓		LCM ↓	LTG ▼		H-OXC ↓		BRV ↓	BRV ↓			RFN ↓			TGB ▼	TPM ▼			ZNS ▼
PHT				CBZ ↓	N-CLB ↑	CNB ↓	ESL ↓	ESM ↓	FBM ▼		LCM ↓	LTG ▼		H-OXC ↓	PER ▼	PHB ↓		PGB ↓	PRM ↓	RFN ↓	STM ↓		TGB ▼	TPM ▼	VPA ↓		ZNS ▼
PGB																											
PRM				CBZ ↓				ESM ↓	FBM ↓		LCM ↓	LTG ▼		OXC ↓			PHT ↓		PRM ↓	RFN ↓	STM ↓		TGB ▼				ZNS ▼
RFN			RFN ▲	CBZ ↓												PH B↑											
STM					CLB ↑																						
STP			7-OH-CBD ▼ 7-COOH-CBD ▼	CBZ ▲	CLB ▲ NCL B▲			ESM ↑								PHB ▲	PHT ▲						TGB ▲		VPA ▲		
TGB																											
TPM			CBD ↓				ESL ↓																				
VPA				CBZ-E ↑				VPA ↓	FBM ↑			LTG ▲								RFN ↑				TPM ↓			
VGB				CBZ ▼					VGB ↑							PHB ↓	PHT▼		PRM ↓	RFN ↓							
ZNS																PHB ↓											

BRV, brivaracetam; CAN, cannabidiol; CBZ, carbamazepine; CBZ-E, carbamazepine-10,11-epoxide (active metabolite of CBZ); CNB, cenobamate; CLB, clobazam; CZP, clonazepam; ESL-a, eslicarbazepine acetate; ESL, eslicarbazepine (active metabolite of ESL-a); ESM, ethosuximide; FBM, felbamate; GBP, gabapentin; H-OXC, 10-hydroxycarbazepine (active metabolite of OXC); LCM, lacosamide; LEV, levetiracetam; LTG, lamotrigine; N-CLB, N-desmethylclobazam (active metabolite of CLB); OXC, oxcarbazepine; PB, phenobarbital; PHT, phenytoin; PGB, pregabalin; PER, perampanel; PRM, primidone; RFN, rufinamide; RTG, retigabine; STM, sulthiame; STP, stiripentol; TGB, tiagabine; TPM, topiramate; VGB, vigabatrin; VPA, valproic acid; ZNS, zonisamide: ↓, a usually minor (or inconsistent) decrease in plasma level; ▼, a usually clinically significant decrease in plasma level; ↑, a usually minor (or inconsistent) increase in plasma level; ▲, a usually clinically significant increase in plasma level; c, h, r–cardiac, hepatic or renal impairment risk, respectively.

**Table 3 ijms-22-09582-t003:** Potentially relevant drug–drug interactions between AEDs and medications used in the treatment of COVID-19 patients [75,76].

	Drugs Used in the Treatment of COVID-19 Patients												
AEDs	ATV	DRV/c	LPV/r	RDV	FAVI	SOF	CLQ	HCLQ	NITA	RBV	TCZ	IFN-β-1a	OSV
BRV	↔	↔	BRV ↓	↔	↔	↔	BRV ↑	BRV ↑	↔	↑BRV	↔	↔	↔
CBD	↔	CBD ↑	CBD ↑	↔	↔	↔	CBD ↑	CBD ↑, HCLQ ↑	↔	↔	↔	↔	↔
CBZ	CBZ ↑, ATV ↓	CBZ ↑, DRV/c ↓	CBZ ↑, LPV/r ↓	RDV↓	↔	SOF↓	CBZ ↓, CLQ ↓	CBZ ↓, HCLQ ↓	↔	↔	CBZ ↓	↔	↔
CLB	CLB ↑	CLB ↑	CLB ↑	↔	↔	↔	↔	↔	↔	↔	↔	↔	↔
CNB	ATV ↓	DRV/c ↓	CNB ↑, LPV/r ↓	↔	↔	↔	CLQ ↓	HCLQ ↓	↔	↔	↔	↔	↔
ESL-a	ATV ↓	DRV/c ↓	LPV/r ↓	RDV ↓	↔	↔	CLQ ↓	HCLQ ↓	↔	↔	↔	↔	↔
ESM	ESM ↑	ESM ↑	ESM ↑	↔	↔	↔	↔	↔	↔	↔	↔	↔	↔
FBM	FBM ↓	DRV/c ↓	FBM ↓	↔	↔	↔	FBM ↓	FBM ↓	↔	↔	↔	↔	↔
GBP	↔	↔	↔	↔	↔	↔	↔	↔	↔	↔	↔	↔	↔
LCM	↔	DRV/c ↑	LCM ↑	↔	↔	↔	LCM ↓	LCM ↓	↔	↔	LCM ↓	↔	↔
LTG	↔	LTG ↑	LTG ↓	↔	↔	↔	LTG ↓	LTG ↓	↔	↔	↔	↔	↔
LEV	↔	↔	↔	↔	↔	↔	↔	↔	↔	↔	↔	↔	↔
OXC	ATV ↓	OXC ↓, DRV/c ↓	LPV/r ↓	RDV ↓	↔	SOF ↓	OXC ↓, CLQ ↓	OXC ↓, HCLQ ↓	↔	↔	↔	↔	↔
PER	PER ↑	DRV/c ↓	PER ↑	↔	↔	↔	↔	↔	↔	↔	PER ↓	↔	↔
PHB	ATV ↓	PHB ↓, DRV/c ↓	LPV/r ↓	RDV ↓	↔	SOF ↓	PHB ↓, CLQ ↓	PHB ↓, HCLQ ↓	↔	↔	PHB ↓	↔	↔
PHT	ATV ↓	DRV/c ↓	PHT ↓, LPV/r ↓	RDV ↓	↔	SOF ↓	PHT ↓, CLQ ↓	PHT ↓, HCLQ ↓	PHT ↑	↔	PHT ↓	↔	↔
PGB	↔	↔	↔	↔	↔	↔	↔	↔	↔	↔	↔	↔	↔
PRM	ATV ↓	DRV/c ↓	PRM ↓, LPV/r ↓	RDV ↓	↔	↔	CLQ ↓	HCLQ ↓	↔	↔	PRM ↓	↔	↔
RFN	ATV ↓	DRV/c ↓	LPV/r ↓	RDV ↓	↔	↔	CLQ ↓	HCLQ ↓	↔	↔	↔	↔	↔
STM	STM ↑	STM ↑	STM ↑	↔	↔	↔	↔	↔	↔	↔	↔	↔	↔
STP	↔	↔	↔	↔	↔	↔	↔	↔	↔	↔	↔	↔	↔
TGB	TGB ↑	TGB ↑	TGB ↑	↔	↔	↔	↔	↔	↔	↔	↔	↔	↔
TPM	↔	DRV/c ↓	LPV/r ↓	↔	↔	↔	↔	↔	↔	↔	↔	↔	↔
VPA	↔	DRV/c ↓	VPA ↓, LPV/r ↑	↔	↔	↔	↔	↔	↔	↔	↔	↔	↔
VGB	↔	↔	↔	↔	↔	↔	↔	↔	↔	↔	↔	↔	↔
ZNS	↔	ZNS ↑	ZNS ↑	↔	↔	↔	↔	↔	↔	↔	↔	↔	↔

AEDs antiepileptic drugs: BRV, brivaracetam; CBD, cannabidiol; CBZ, carbamazepine; CBZ-E, carbamazepine-10,11-epoxide (active metabolite of CBZ); CNB, cenobamate; CLB, clobazam; ESL-a, eslicarbazepine acetate; ESM, ethosuximide; FBM, felbamate; GBP, gabapentin; LCM, lacosamide; LEV, levetiracetam; LTG, lamotrigine; OXC, oxcarbazepine; PB, phenobarbital; PHT, phenytoin; PGB, pregabalin; PER, perampanel; PRM, primidone; RFN, rufinamide; RTG, retigabine; STM, sulthiame; STP, stiripentol; TGB, tiagabine; TPM, topiramate; VGB, vigabatrin; VPA, valproic acid; ZNS, zonisamide. COVID-19 drugs: ATV, atazavir; DRV/c, darunavir/cobicistat; LPV/r, lopinavir/ritonavir; RDV, remdesevir/GS-5734; FAVI, favipiravir; SOF, sofosbuvir; CLQ, chloroquine; HCLQ, hydroxychloroquine; NITA, nitazoxanide; RBV, ribavirin; TCZ, toclizumab; IFN-β-1a, interferon β-1a; OSV, oseltamivir; ↑, potential increased exposure; ↓, potential decreased exposure; ↔, no significant effect.

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
