# Peer review of "Pharmacokinetic Drug–Drug Interactions among Antiepileptic Drugs, Including CBD, Drugs Used to Treat COVID-19 and Nutrients"

_ijms, 2021, doi:10.3390/ijms22179582_

Round 1
Reviewer 1 Report
The introduction section has to be improved: there is lack of informations about cannabidiol.
A paragraph describing the main features of cannabidiol could be also appropriate.
Line 254: "Grapefruit juice increases the bioavailability of CBZ by inhibiting CYP3A4 enzymes in the gut wall and in the liver": Please include appropriate reference.
Author Response
We are grateful for your critical comments. Based on the suggestions, we have made careful modifications to the original manuscript. The point-to-point replies and explanations for all of the revisions are listed below.
1) The introduction section has to be improved: there is lack of informations about cannabidiol.
Answer: Information on cannabidiol was added to the introduction as suggested.
2) A paragraph describing the main features of cannabidiol could be also appropriate.
Answer: Section “2. Cannabidiol” was introduced to the manuscript. The section contains information on cannabidiol, including its pharmacology, pharmacokinetics and drug-drug interactions with AEDs and other drugs, such as medications used in the treatment of patients with COVID-19.
3) Line 254: "Grapefruit juice increases the bioavailability of CBZ by inhibiting CYP3A4 enzymes in the gut wall and in the liver": Please include appropriate reference.
Answer: The following reference was added containing the newest information on the subject:
Karmakar, S.; Biswas, S.; Bera, R.; Mondal, S.; Kundu, A.; Ali, M.A.; Sen, T. Beverage-Induced Enhanced Bioavailability of Carbamazepine and Its Consequent Effect on Antiepileptic Activity and Toxicity. Journal of Food and Drug Analysis 2015, 23, 327–334, doi:10.1016/j.jfda.2014.07.012.
Reviewer 2 Report
This paper is aimed to systematically review the pharmacokinetic interactions among antiepileptic drugs, but also among antiepileptic drugs and ketogenic diet and finally is focused on the potential interactions among antiepileptics and drugs used for Covid 19 infection treatment.
For each drug a brief history is provided, pharmacokinetic information and metabolism are illustrated and potentially interaction with the other antiepileptic drugs examined. Some paragraphs are then dedicated to the interaction with ketogenic diet and with drugs used for Covid 19 infection treatment, in particular antiviral drugs.
There are many reviews in literature which examined drug-drug interactions among antiepileptics and also among antiepileptics and cannabinoids. Many papers were also published on the interaction between antiepileptics and ketogenic diet. The paper seems to me not innovative in this regard.
The new topic is the interaction among antiepileptics and drugs used for Covid 19 infection treatment. Therefore, in my opinion the paper should be focused on this aspect.
The general presentation of each drug could be relatively condensed. Information among pharmacokinetics and metabolism of the different drugs should be summarized in a table. In addition the table illustrating expected changes in plasma concentrations when an antiepileptic drug (AED) is added to a pre-existing AED regimen should be introduced only to emphasize that antiepileptic polytherapy further complicate the management of epileptic patients with Covid 19 infection.
Conversely the section dedicated to the interaction among each antiepileptic and drugs used for Covid 19 treatment should be more detailed and if possible presented in a summary table.
The Conclusions should be rewritten according the more focused aim of the review.
Author Response
We are grateful to the Reviewer for all critical comments. Based on these suggestions, we have made careful modifications to the original manuscript. The point-to-point replies and explanations for all of the revisions are listed below.
This paper is aimed to systematically review the pharmacokinetic interactions among antiepileptic drugs, but also among antiepileptic drugs and ketogenic diet and finally is focused on the potential interactions among antiepileptics and drugs used for Covid 19 infection treatment.
For each drug a brief history is provided, pharmacokinetic information and metabolism are illustrated and potentially interaction with the other antiepileptic drugs examined. Some paragraphs are then dedicated to the interaction with ketogenic diet and with drugs used for Covid 19 infection treatment, in particular antiviral drugs.
There are many reviews in literature which examined drug-drug interactions among antiepileptics and also among antiepileptics and cannabinoids. Many papers were also published on the interaction between antiepileptics and ketogenic diet. The paper seems to me not innovative in this regard.
The new topic is the interaction among antiepileptics and drugs used for Covid 19 infection treatment. Therefore, in my opinion the paper should be focused on this aspect.
Answer: The reports on clinically significant interactions between AEDs and drugs used for Covid 19 infection treatment are scarce. Therefore, we decided not to change the main topic and structure of our article. However, we have tried to add all avaiable details regarding potential interaction between AEDs and medications used in the treatment of COVID-19 infection. Moreover, cardiac, hepatic, or renal problems, which may happen in patients with severe COVID-19 infection, and may require adjustment to AEDs were also reported.
The general presentation of each drug could be relatively condensed. Information among pharmacokinetics and metabolism of the different drugs should be summarized in a table. In addition the table illustrating expected changes in plasma concentrations when an antiepileptic drug (AED) is added to a pre-existing AED regimen should be introduced only to emphasize that antiepileptic polytherapy further complicate the management of epileptic patients with Covid 19 infection.
Answer: We have condensed the general presentation of each drug as suggested. Information on pharmacokinetics including metabolism have been presented in a table (Table 1). The table illustrating expected changes in plasma concentrations when an antiepileptic drug (AED) is added to a pre-existing AED regimen (Table 2), was supplemented with the information about possible cardiac, renal and hepatic complications which may complicate the management of epileptic patients with Covid 19 infection.
Conversely the section dedicated to the interaction among each antiepileptic and drugs used for Covid 19 treatment should be more detailed and if possible presented in a summary table.
Answer: We have added more details regarding the interaction between each antiepileptic drug and medications used for COVID-19 treatment. Moreover, the interactions were presented in a summary table (Table 3) as suggested.
The Conclusions should be rewritten according the more focused aim of the review.
Answer: The conclusion section regarding interaction between AEDs and anti-COVID drugs was supplemented with additional information. Challenging issues that may happen in epileptic patients, who have COVID-19 and are receiving treatment with AEDs were underlined. Some tips were added that might be useful for clinicians.